# Arginine-Containing Tripeptides as Analgesic Substances: The Possible Mechanism of Ligand-Receptor Binding to the Slow Sodium Channel

**DOI:** 10.3390/ijms23115993

**Published:** 2022-05-26

**Authors:** Ilya V. Rogachevskii, Vera B. Plakhova, Valentina A. Penniyaynen, Arina D. Kalinina, Svetlana A. Podzorova, Dmitriy M. Samosvat, Georgy G. Zegrya, Boris V. Krylov

**Affiliations:** 1Pavlov Institute of Physiology of the Russian Academy of Sciences, 199034 Saint Petersburg, Russia; roggie_spb_ru@yahoo.com (I.V.R.); verapl@mail.ru (V.B.P.); pvalentina2@yandex.ru (V.A.P.); arinakalinina95@gmail.com (A.D.K.); podzorova@infran.ru (S.A.P.); 2Ioffe Institute of the Russian Academy of Sciences, 194021 Saint Petersburg, Russia; samosvat@yandex.ru (D.M.S.); zegrya@theory.ioffe.ru (G.G.Z.)

**Keywords:** arginine-containing tripeptides, Na_V_1.8 channel, patch-clamp method, conformational analysis, nociception, analgesics

## Abstract

Two short arginine-containing tripeptides, H-Arg-Arg-Arg-OH (TP1) and Ac-Arg-Arg-Arg-NH_2_ (TP2), have been shown by the patch-clamp method to modulate the Na_V_1.8 channels of DRG primary sensory neurons, which are responsible for the generation of nociceptive signals. Conformational analysis of the tripeptides indicates that the key role in the ligand-receptor binding of TP1 and TP2 to the Na_V_1.8 channel is played by two positively charged guanidinium groups of the arginine side chains located at the characteristic distance of ~9 Å from each other. The tripeptide effect on the Na_V_1.8 channel activation gating device has been retained when the N- and C-terminal groups of TP1 were structurally modified to TP2 to protect the attacking peptide from proteolytic cleavage by exopeptidases during its delivery to the molecular target, the Na_V_1.8 channel. As demonstrated by the organotypic tissue culture method, the agents do not affect the DRG neurite growth, which makes it possible to expect the absence of adverse side effects at the tissue level upon administration of TP1 and TP2. The data obtained indicate that both tripeptides can have great therapeutic potential as novel analgesic medicinal substances.

## 1. Introduction

The objective of the current study is to elucidate the molecular mechanism of ligand-receptor binding of short arginine-containing peptides to the slow sodium Na_V_1.8 channel that is responsible for the primary sensory coding of nociceptive signals [1]. These channels can be considered markers of dorsal root ganglia (DRG) nociceptive neurons [2]. Relatively low-frequency impulse firing of mechano-, chemo-, and thermoreceptors conveys to the CNS the information about an adequate stimulus, whereas excessive and/or damaging stimulation evokes an increase in impulse firing frequency informing the CNS about noxious stimuli [3]. Specific ligand-receptor binding of a number of agents (comenic acid and ouabain applied at nanomolar concentrations) to the corresponding membrane targets has been demonstrated to decrease voltage sensitivity of the Na_V_1.8 channel activation gating device. It has been shown to result in a selective decrease in the high-frequency component of afferent nociceptor impulse firing and, ultimately, in peripheral pain relief [4,5]. Analog-impulse signal conversion performed by tissue nociceptors is thus delicately modulated at the molecular level [6].

Several arginine-containing short peptides have also been shown to modulate the Na_V_1.8 channel activation gating device [7,8]. The peptides are suggested to bind directly to the Na_V_1.8 channel due to intermolecular ion–ion bonding between positively charged guanidinium groups of the arginine side chains and nucleophilic moieties of the Na_V_1.8 channel molecule. Hence, it is a reasonable assumption that the distance between guanidinium groups should match certain criteria to provide electrostatic and steric complementarity for effective ligand-receptor binding [8].

The free arginine molecule (R) and the Ac-RR-NH_2_ dipeptide have not been found to control voltage sensitivity of the Na_V_1.8 channel activation gating device [7], which makes the RRR sequence the shortest potentially functional peptide sequence that includes only arginine residues. Therefore, an investigation of its effect on the Na_V_1.8 channel is expected to provide evidence in support of the suggestion that only guanidinium groups of arginines are absolutely required for effective ligand-receptor binding of arginine-containing peptides.

The endogenous nature of short peptides makes them promising candidates for safe clinical application because adverse side effects are mainly triggered by more structurally sophisticated polypeptide molecules that inhibit their target membrane receptors rather than modulate them. To develop a novel peptide medicinal substance, the challenging problem of its delivery to the molecular target avoiding proteolytic decomposition has to be solved [9].

Structural modification, such as N-terminal acetylation and C-terminal amidation, can protect a peptide from cleavage by exopeptidases. However, another chemically individual molecule that is thus obtained does not a priori retain the pharmacological effect of the native peptide, which is even more questionable if the peptide is short. At the very least, ionized N-terminal amino group and C-terminal carboxylate are correspondingly modified into neutral, considerably bulky acetylamino groups and carboxamides. Such a conversion inevitably alters the electrostatic and hydrophobic properties of the entire molecule and might ultimately modulate the efficacy of its binding to the membrane target.

In order to elucidate whether the introduction of C- and N-terminal protection groups necessarily affects the ability of short arginine-containing peptides to modulate the Na_V_1.8 channel activation gating device (the key effect for their application as medicinal substances), the tripeptides H-RRR-OH (TP1) and Ac-RRR-NH2 (TP2) were investigated.

## 2. Results

### 2.1. The Patch-Clamp Method

Families of Na_V_1.8 sodium currents recorded in the control experiment and after extracellular application of TP1 and TP2 at 100 nM are shown in Figure 1a,b. Normalized peak current-voltage characteristics of sodium currents plotted using the regular protocol [10] demonstrate a slight shift of the left branch along the voltage axis, which indicates that both tripeptides modulate voltage sensitivity of the Na_V_1.8 channel activation gating device (Figure 1c,d).

The effective charge of the Na_V_1.8 channel activation gating device (Z_eff_) is an important quantitative parameter that determines the voltage sensitivity of the nociceptive neuron membrane. To evaluate it, the dependences of the chord conductance on transmembrane potential difference G_Na_(E) were plotted based on experimental recordings. The G_Na_(E) function has an initial S-shaped segment, the slope of which describes the distinctive features of voltage sensitivity of the activation process. The changes in this slope resulting from the application of both TP1 and TP2 become more clearly observed when normalized G_Na_^norm^(E) functions are constructed (Figure 2a,b).

The Boltzmann distribution is regularly used to describe the G_Na_(E) function. The stationary characteristics of transitions between the states of the Na_V_1.8 channel activation gating device are assumed in this case to be determined by the voltage dependence of the chord conductance. To obtain the Z_eff_ values herein, we have implemented a different approach, first suggested by the authors of the membrane ionic theory [10] and later modified [11]. The protocol of Z_eff_ evaluation is illustrated in Figure 2c,d. The tangents of the slopes of the asymptotes passing through the first three points of the L(E) function determine the limiting logarithmic sensitivity of the Na_V_1.8 channel to transmembrane potential change [11]. The Z_eff_ value decreased from 6.1 electron charge units in the control experiment to 4.7 electron charge units after the application of TP1 (Figure 2c) and from 6.8 to 4.8 after the application of TP2 (Figure 2d). Data presented in Figure 3 clearly indicate a significant decrease in Z_eff_ after the application of either tripeptide. Experimental data recorded on approximately 20 neurons in each case made it possible to obtain the averaged Z_eff_ values in control experiments and after TP1 and TP2 application.

According to the Gouy–Chapman–Stern theory, the sodium gating mechanism is sensitive to the local transmembrane potential, which is different from the bulk-to-bulk membrane potential due to the surface charges at the external membrane [12,13]. Ligand-receptor binding of the attacking tripeptide modulates the Na_V_1.8 channel activation gating device and, therefore, the local electric field, which should shift the L(E) function in the depolarizing direction along the voltage axis. Unfortunately, only a qualitative explanation of the observed shift can be provided because the detailed mechanism of molecular events triggered by ligand-receptor binding of the tripeptides to the Na_V_1.8 channel, ultimately resulting in the modulation of its activation gating device, is not yet clear.

### 2.2. Organotypic Tissue Culture

During the first day of DRG explants culturing, the explants spread over the collagen substrate; neurite growth and eviction of proliferating and migrating cells began. After 3 days of culturing, 2 zones can be distinguished, both in the control and experimental DRG explants. The peripheral zone (also called the growth zone) contains fibroblast-like cells, glia, single migrating neurons, and a neuronal network composed of long, thick, thin, and ramified neurites, while the central zone consists of nonmigrating differentiating neuroblasts. Examples of control, TP1-, and TP2-treated DRG explants registered and visualized by confocal laser microscopy do not display a distinctive difference (Figure 4a–c). No significant effect of either tripeptide on the area index (AI) and average neurite length values in DRG explants was detected in a wide range of applied concentrations, from 100 pM to 10 μM (Figure 5a,b). Obtained data indicate that TP1 and TP2 do not modulate DRG neurite growth, and adverse side effects should not be expected at the tissue level upon their medicinal application.

### 2.3. Calculational Methods

The lowest energy conformations of the tripeptides are shown in Figure 6. TP1 was considered in the zwitterion form, while the N- and C-terminal groups of TP2 were uncharged. Average distances between the guanidinium groups in TP1 and TP2 calculated over the entire ensemble and smaller subensembles are presented in Table 1. Relatively large SEM values are indicative of a high degree of conformational freedom in both molecules, the overall conformations of which should be mainly determined by electrostatic repulsion of three positively charged guanidinium groups.

Monitoring changes in the mean values of distances between guanidinium groups in correlation with a gradual decrement of the energy cutoff defining the amount of lowest energy conformations to be included in a subensemble can be considered as a way to perform a numerical limiting process within the ensemble of conformations. The R^1^–R^2^ and R^2^–R^3^ distance values demonstrate a stable behavior, especially the former, which remains fairly similar in all calculations. A certain decrease in the R^2^–R^3^ distance in TP1 reflects the stabilization of its lowest energy conformations by intramolecular salt bonding between a guanidinium group and C-terminal carboxylate. As it seems to be, the R^3^ guanidinium group is predominantly involved because the R^1^–R^2^ distance value should be affected otherwise. There is no such opportunity in TP2, as its C-terminal carboxyl group is amidated and uncharged. Average distances between guanidinium groups in TP2 calculated over the entire ensemble are equal, which supports the suggestion that the overall tripeptide conformation is controlled by electrostatic repulsion of three positive charges. They form an almost ideal equilateral triangle, somewhat distorted in TP1 due to intramolecular salt bonding missing in TP2.

A rather nontrivial result is the observed tendency for the R^1^–R^3^ distance to smoothly decrease from the value of ~11 Å with the decrement of the energy cutoff. The R^1^–R^3^ distance is the only parameter in both molecules which depends on the energies of conformations included in the subensemble for averaging. It is reasonable to assume that a reliable subensemble should include at least 1% of the total count of conformations, around 1000, to avoid inadequate sampling of conformational space. Following this logic, the limiting value of R^1^-R^3^ distance (~9 Å) is obtained with the energy cutoff of 4.5 kcal/mol.

We suggest that ligand-receptor binding of TP1 and TP2 to the Na_V_1.8 channel necessarily involves two guanidinium groups of the R^1^ and R^3^ arginine side chains with the characteristic R^1^-R^3^ distance of ~9 Å. Because both tripeptides are shown to modulate the effective charge of the Na_V_1.8 channel activation gating device, the N- and C-terminal functional groups do not seem to play a fundamental role in the binding process. 

## 3. Discussion

Using the complicated method of displacement currents recordings, Armstrong and Bezanilla (1973) [14], for the first time experimentally, recorded a nonlinear capacitive current of the squid axon membrane, which they associated with the charge displacement in the activation gating device of sodium channels. According to the modern classification, these were Na_V_1.1 channels.

The main disadvantage of the displacement currents method for evaluating charge transfer of the sodium channel activation gating device is that the cell membrane contains many other proteins, the charged components of which can reduce the accuracy of measuring this characteristic of the sodium channel. 

The progress in measuring the charge transferred by opening the sodium channel activation gate was made by Almers (1978) [11], who developed a fundamentally new approach that was free from the above limitations. However, unfortunately, the Almers’ method has other limitations, which make it impossible to measure the gating charge transfer of many ion channels, including Na_V_1.1. This is because the method is valid only if the behavior of the activation gating device obeys the case of a linear sequence of discrete closed states followed by only one open state. At the same time, other gating processes should not be superimposed on the activation gate opening.

In accordance with our hypothesis, it is the slow sodium Na_V_1.8 channel that is perfectly suitable for applying the Almers’ method to study the activation gating device charge transfer for the following reasons. Slow sodium channels earned their name due to the fact that the processes of activation (fast) and inactivation (slow) of the sodium current are separated in time. In other words, the peak (amplitude) value of the sodium current is reached at the moment when the inactivation process does not yet affect this value. This situation is fundamentally different from the behavior of the Na_V_1.1 inactivation system, for which the Almers’ method is inapplicable [1]. 

The principal Almers’ theory requirement of a single transition between one closed and one open state of the activation gating device is not violated for Na_V_1.8 channels. The methodological approach applied herein allows us to selectively measure the peak current–voltage characteristics associated exclusively with the functioning of the Na_V_1.8 channels and further construct the chord conductivity voltage dependence. The limiting slope method introduced by Almers exploits the relationship between the charge movement and voltage sensitivity of the chord conductance, yielding the value of effective charge transferred by a single channel gating device. 

In our practice, this technique does not suffer from low experimental resolution for two reasons. First, the application of the patch-clamp method greatly increases the accuracy of the approach. Second, we use microelectrodes with low series resistance. Theoretical analysis of the accuracy of the patch-clamp method has demonstrated that the static and dynamic errors of the method can be thus minimized [15]. As a result, it becomes possible to register weak nonlinearities that manifest themselves only when the Almers’ function L(E) approaches the limit at the most negative values of the membrane potential E. It is in this case that the logarithm of the L(E) function becomes a linear dependence, and the tangent of its slope carries information about the parameter that has a physical meaning not only as an effective charge of the Na_V_1.8 activation gating device but also as a physiological quantitative indicator of its voltage sensitivity.

It is important to emphasize that the discovery of weak nonlinearities, which we have used earlier to describe the sodium channel gating devices that do not follow the simple one-barrier model, led to the finding of new molecular mechanisms accounting for relevant physiological phenomena such as adaptation [16]. In the present work, the application of this approach has also made it possible to correlate the results of the theoretical conformational analysis with the experimentally obtained data on the sodium currents in Na_V_1.8 channels. The only significant drawback of the applied approach is the impossibility of using standard statistical methods. The fact is that the studied nonlinearities are really weak; therefore, statistical averaging at an early stage of data processing completely hides the component that is the goal of our research. The situation is aggravated by the run-down effect that is inherent to the patch-clamp method and manifests itself in a decrease in the sodium current amplitudes during the experiment. Our use of electrodes with low input resistance enhances this effect due to the replacement of the intracellular environment of the neuron, which occurs differently in different cells. We have nevertheless managed to reliably isolate and register the Almers’ L(E) function on each individual selected neuron both in control experiments and after application of the TP1 (H-RRR-OH) and TP2 (Ac-RRR-NH_2_) tripeptides. Effects of the tripeptides on effective charge transfer are clearly seen. Whether TP1 and TP2 affect the density of Na_V_1.8 channels in the nociceptive neuron membrane remains an open question. The run-down effect makes it impossible to answer this question with sufficient accuracy when randomizing our experiments. In our further work, the very sensitive immunofluorescence method will be applied to detect not only the modulating but also the blocking effects of the tripeptides, as it has been done before in the study of ouabain [17].

The principal result of the current work is the in vitro identification of equally potent effects of TP1 and TP2 on the Na_V_1.8 channel activation gating device. A decrease in the effective charge most likely results from the binding of either peptide to a so-far unidentified target site on the Na_V_1.8 channel molecule. The ability of the tripeptides to modulate voltage sensitivity of the Na_V_1.8 channel is highly specific, as the concentrations of TP1 and TP2 evoking a significant decrease in Z_eff_ are in the nanomolar range (100 nM). Experimental data strongly suggest that both molecules are promising candidates for the role of a novel analgesic medicinal substance because our prior experience with respect to the receptor-mediated effect of comenic acid demonstrates a correlation between the specific decrease in Z_eff_ and antinociceptive effect at the organismal level [1,4,5,18,19].

No effect of TP1 and TP2 on DRG neurite growth was detected. It indicates that the tripeptides, as opposed to earlier studied agents (comenic acid and ouabain applied at nanomolar concentrations), do not trigger the receptor- or transducer-mediated signaling cascades, the activation of which modulates the Na_V_1.8 channel activation gating device [5,17]. This fact supports our suggestion that TP1 and TP2 bind directly to the Na_V_1.8 channel due to intermolecular ion–ion bonding between positively charged guanidinium groups of the attacking peptide molecule and nucleophilic moieties of the Na_V_1.8 channel. This suggestion has been put forward earlier when the effects of the Ac-RER-NH_2_ tripeptide and the Ac-RERR-NH_2_ tetrapeptide were investigated [7,8]. However, the aforementioned peptides contain glutamic acid residue (E). It could not thus be unambiguously concluded that guanidinium groups of the arginine side chains were specifically responsible for the ligand-receptor binding of the peptides to the Na_V_1.8 channel, which resulted in a decrease in the effective charge of its activation gating device.

Calculational data demonstrate that the R^1^-R^3^ distance is the only distance between guanidinium groups in both molecules, the value of which depends on the energy cutoff determining the amount of the lowest energy conformations included in the subensemble for averaging. It suggests that guanidinium groups of the R^1^ and R^3^ side chains are directly involved in intramolecular ion–ion bonding upon ligand-receptor binding of the tripeptides, and the characteristic distance between guanidinium groups in the molecules of arginine-containing peptides required to provide steric and electrostatic ligand-receptor complementarity for binding to the Na_V_1.8 channel is 9 Å.

The TP1 and TP2 tripeptides were intentionally designed to contain only guanidinium functional groups as potential pharmacophores if the N- and C-terminal groups are taken out of consideration. To address how the chemical nature of these groups might modulate the physiological effect of short arginine-containing peptides, structural modification of both terminal groups in TP1 has been carried out to obtain TP2. The zwitterion TP1 tripeptide has thus been modified into the N-acetylated and C-amidated TP2 with uncharged terminal groups. 

Presently obtained data demonstrate that effective ligand-receptor binding of short arginine-containing peptides to the Na_V_1.8 channel requires only positively charged guanidinium functional groups. It is important to stress, once again, that these groups should be located at a characteristic distance from each other for ligand-receptor binding to occur. Our data suggest that this distance is about 9 Å. According to our hypothesis, the larger the distance between two guanidinium groups deviates from the characteristic value, the lower the chances are that the given two groups are simultaneously required for binding to the Na_V_1.8 channel. However, electrostatic and steric complementarity of the peptide to its binding site is only a necessary condition to consider it as a potentially medicinal substance. Another issue to be resolved is the effective in vivo delivery of the agent to its molecular membrane target, which might require structural modification of the N- and C-terminal groups to avoid possible cleavage of the peptide molecule by exopeptidases [9]. It should, though, be taken into account that such a modification might affect the peptide conformation, and the resulting distance between the pharmacophore guanidinium groups might, therefore, not correspond to the characteristic value.

If successfully delivered to their membrane target, the slow sodium Na_V_1.8 channel, both TP1 and TP2 tripeptides are promising candidates for the role of a novel analgesic medicinal substance. Structural modification of the N- and C-terminal groups of TP1 to TP2 has not been shown to affect the ability to decrease the effective charge of the Na_V_1.8 channel activation gating device and significantly change the distance between the guanidinium groups directly involved in ligand-receptor binding of the peptides. The combined application of the patch-clamp method, organotypic tissue culture method, and conformational analysis is a challenging methodology to study the effects of short arginine-containing peptides on the Na_V_1.8 channel for the development of novel analgesic substances. This combination of methods allows us to investigate within a single study the target physiological effect of the attacking molecule, its possible adverse side effects on the nerve tissue, and obtain the necessary structural information to elucidate the mechanism of its ligand-receptor binding.

## 4. Materials and Methods

### 4.1. Chemicals and Reagents

All chemicals, excluding TP1 and TP2, were purchased from Sigma (Sigma-Aldrich, St. Louis, MA, USA). Short peptides TP1 and TP2 were synthesized in Verta Research and Production Company (St. Petersburg, Russia) by the method of classic peptide synthesis using reagents from Sigma (Sigma-Aldrich, St. Louis, MA, USA) and Iris Biotech GmbH (Marktredwitz, Germany) and characterized with high-performance liquid chromatography (purity of more than 95%) and mass spectrometry.

### 4.2. Patch-Clamp Method

#### 4.2.1. Dissociated Cell Culture

Experiments were performed on dissociated sensory neurons using a short-term cell culture technique. DRG isolated from the L5-S1 region of the spinal cord of newborn Wistar rats was placed in Hank’s solution. The time of enzymatic treatment varied from 2 to 5 min at 37 °C [20]. The solution containing 1 mL Hank’s solution, 1 mL Eagle’s medium, 2 mg/mL type 1A collagenase, 1 mg/mL pronase E, and 1 mM HEPES Na was used, pH = 7.4. Given the applied enzyme concentrations, the treatment time was chosen so as to optimally provide a sufficient amount of viable neurons. Centrifugation (1 min, 900 rpm) was carried out after enzymatic treatment and thorough washing by changing the supernatant solution several times. Washing and subsequent culturing were performed in Eagle’s medium with the addition of fetal bovine serum (FBS, 10%), glucose (0.6%), gentamicin (40 U/mL), and glutamine (2 mM). Isolated neurons were further obtained by mechanical dissociation by pipetting. Non-neuronal elements (Schwann cells and fibroblasts) were allowed to settle on the bottom of a plastic 90-mm Petri dish for 25 min at 37 °C, which made it possible to obtain the cell culture consisting predominantly of dissociated sensory neurons. The neurons were then cultured in collagen-coated 40-mm Petri dishes. Visually unimpaired cells were chosen for further experiments after 1–2 h of culturing. Ionic currents were recorded on the cells, which remained viable for several hours.

#### 4.2.2. Experimental Solutions 

Slow sodium Na_V_1.8 currents were investigated using the following solutions. The extracellular solution contained 70 mM choline chloride, 65 mM NaCl, 10 mM HEPES Na, 2 mM CaCl_2_, 2 mM MgCl_2_, and 0.1 µM tetrodotoxin, pH = 7.4. The extracellular sodium level was chosen at half of its normal concentration, which resulted in a decrease in the amplitude of the Na_V_1.8 current. Correspondingly, this effect increased the accuracy of measurements, as the stationary error caused by series resistance (see below) is thus lower. The intracellular solution contained 100 mM CsF, 40 mM CsCl, 10 mM NaCl, 10 mM HEPES Na, and 2 mM MgCl_2_, pH = 7.2. The pH values were adjusted with HCl. The composition of extra- and intracellular solutions allowed the elimination of all potassium currents due to the absence of potassium ions, while intracellular fluoride ions blocked the calcium currents [21]. Tetrodotoxin was added to the extracellular solution to block all fast tetrodotoxin-sensitive sodium channels, which made it possible to register the responses of slow tetrodotoxin-resistant Na_V_1.8 sodium channels only. When responses of other slower tetrodotoxin-resistant sodium channels were visually detected in the recordings of ionic currents, the experiment was terminated. Single neurons were placed in the experimental bath (volume 200 μL) using a micropipette. The external solution in the bath was refreshed by passive flow under gravity. 

#### 4.2.3. Hardware and Software 

The patch-clamp method was utilized in the «whole-cell recording» configuration [22] using the hardware–software set-up, which involved the patch-clamp L/M-EPC 7 amplifier, digital–analog and analog–digital converters, and the computer. The software package for the automated running of experiments was developed in the Laboratory of Physiology of Excitable Membranes of Pavlov Institute of Physiology RAS. Before ionic currents were recorded, intracellular perfusion of the neuron with the microelectrode filling solution was allowed for several minutes. The maximal experiment duration in the «whole-cell recording» configuration was 60 min. 

The software package utilizes the following scheme. In the first place, the sequence of electrical stimuli to be applied to the neuron, that is, the sequence of rectangular voltage steps of different amplitudes, is set. The sodium currents arising in response to this sequence are registered and visually controlled. After the application of voltage steps in accordance with the chosen program, express processing of the neuronal membrane responses is performed to plot the current-voltage curve and the voltage dependence of chord conductance of the sodium currents. The next step of data processing is the construction of the logarithmic voltage sensitivity L(E) using the Almers’ method [11]. It allows us to evaluate the effective charge value (Z_eff_, in electric charge units) transferred by the Na_V_1.8 channel activation gating device. 

The series resistance (R_S_), which determines both dynamic and stationary errors of the patch-clamp method, can be evaluated automatically during the experiment. Its value should not exceed 3 MOhm. Stationary and kinetic parameters of the currents are otherwise obtained with large errors, as demonstrated by theoretical analysis of patch-clamp applicability limitations [15]. Another error is associated with the accuracy of the evaluation of the transmembrane voltage difference (E), which is also determined by R_S_. The actual E value defines the position of stationary characteristics of the Na_V_1.8 channel with respect to the voltage axis. The stationary patch-clamp error can be easily estimated from the expression ΔE = I_Na_^max^R_S_, where I_Na_^max^ is the maximal amplitude of the sodium current family. The actual voltage dependences of stationary functions of Na_V_1.8 channels should be shifted by ΔE to the right along the E axis. During the experiment, small uncontrolled changes in the value of R_S_ can occur, which sometimes, in turn, can lead to a slight shift in the current-voltage functions. If I_Na_^max^ does not exceed 1 nA, the stationary error is less than 1 mV and can be neglected. In addition to R_S_, the membrane capacitance (C_m_) was also evaluated. Both capacitive (I_C_) and leakage (I_L_) currents were subtracted automatically. The peak current-voltage relationship was constructed, making it possible to obtain the chord conductance (G_Na_) and, further, the Z_eff_ value of the Na_V_1.8 channel activation gating device. 

The program of voltage impulses applied to obtain the peak current-voltage relationship included the following steps. The first impulse was equal to the resting potential of −60 mV. The step of holding potential usually equal to −110 mV was generated after. The set of sequential test impulses of 50 ms duration with an increment of 5 mV was used further to record the family of sodium currents in the voltage range from −60 to 45 mV. Registration of the amplitude (peak) values of the currents generated in response to each voltage step made it possible to build the peak current-voltage curve. The reversal potential (E_r_) was also determined as the point of intersection of the right branch of the peak current-voltage function with the voltage axis. 

#### 4.2.4. The Limiting–Slope Procedure

The limiting-slope procedure [11] is the approach used here to estimate Z_eff_ [1]. The ratio of the number of open Na_V_1.8 channels (N_o_) to the number of closed channels (N_c_) is calculated as
N_o_/N_c_ = G_Na_(E)/[G_Na_^max^ − G_Na_(E)]
where G_Na_^max^ and G_Na_(E) are the maximal value and the voltage dependence of the chord conductance, respectively. G_Na_(E) can be obtained in patch-clamp experiments as
G_Na_(E) = I_peak_(E)/(E − E_Na_)
where I_peak_ is the amplitude value of the sodium current and E_Na_ is the reversal potential for sodium ions. G_Na_(E) is a monotonous function approaching its maximum G_Na_^max^ at positive E. According to the Almers’ theory, the limiting-slope procedure can be applied:lim(N_o_/N_c_) = lim{G_Na_(E)/[G_Na_^max^ − G_Na_(E)]} → C∙exp[(Z_eff_
**e**_0_ E)/(kT)]
 E→ −∞ E→ −∞    E→ −∞        
where k is the Boltzmann constant, T is the absolute temperature, C is a constant, and e_0_ is the electron charge.

The practical application of the Almers’ method is as follows. When the membrane potential E approaches minus infinity (E→ −∞), Z_eff_ can be estimated from the slope of the asymptote passing through the first experimental points determined by the very negative values of E, since the Boltzmann’s principle is applicable at these potentials. The Almers’ limiting-slope procedure makes it possible to experimentally evaluate Z_eff_ by constructing the voltage dependence of the logarithmic voltage sensitivity function L(E):L(E) = ln (G_Na_(E)/(G_Na_^max^ − G_Na_(E)).

The asymptote passing through the first points of the L(E) function obtained at the most negative E allows us to calculate Z_eff_, which is linearly proportional to the tangent of the asymptote slope.

### 4.3. Organotypic Nerve Tissue Culture Method

Experiments were performed on DRG explants from 10–12 days old chicken embryos (E10–E12) cultured in collagen-coated 40-mm Petri dishes in the CO_2_-incubator (Sanyo, Japan) for 3 days at 37 °C with 5% CO_2_. The culturing medium contained 45% Hank’s solution, 40% Eagle’s minimal essential medium, 10% FBS with addition of insulin (0.5 U/mL), glucose (0.6%), L-glutamine (2 μM), and gentamicin (100 U/mL). Explants cultured exclusively in the culturing medium served as the control. At 3 days of culture age, the neurite growth was assessed by the morphometric method. This experimental technique is as highly sensitive as the patch-clamp method, which has been demonstrated in our prior studies [4,5,17]. The area index (AI) was calculated as the ratio of the peripheral growth zone area to the central zone area [4]. DRG neurite length was measured after immunocytochemical staining of DRG explants. The growth zone area was divided into six equal sections. The neurite length was measured from the ganglion edge to the edge of the peripheral growth zone for each section. The average neurite length was used to calculate the extent of neurite outgrowth. The Axio Observer Z1 microscope (Carl Zeiss, Germany) was implemented to visualize the objects. Obtained images were analyzed with ImageJ and ZEN_2012 software. Experiments were conducted using the equipment of the Confocal Microscopy Collective Use Center at Pavlov Institute of Physiology RAS.

### 4.4. Immunostaining

Petri dishes containing the DRG explants were rinsed quickly once with phosphate-buffered saline (PBS), fixed using 4% paraformaldehyde for 3 min, washed 3 times with PBS, incubated with PBS containing 0.3% Triton-X for 15 min, washed 3 times with PBS, and blocked by PBS containing 10% FBS for 30 min. DRG explants were then incubated with anti-neurofilament 200 primary antibody overnight at 4 °C, washed 3 times with PBS, incubated with TRIC-conjugated secondary antibody for 2 h, washed again 3 times with PBS, incubated with DAPI for 15 min, washed once with PBS, and stored in PBS at 4 °C for imaging. All the incubations were carried out at room temperature.

### 4.5. Calculational Methods

Conformational analysis was performed using the TINKER 8.0 program package [23] with the MMFF94 force field [24]. Low-mode conformational search method (LMOD) was applied [25] with approximately 100,000 single searches for each structure. Solvation effects were taken into implicit account in the framework of the GB/SA approach [26] with the dielectric constant ε = 10.0 (models dielectric properties of the surrounding milieu upon ligand-receptor binding of the peptide to the Na_V_1.8 channel) and ε = 80.4 (aqueous solution). Guanidinium groups of the arginine side chains (R) were always positively charged, as they were shown to be protonated even in the bulk of the protein [27]. The net charges of TP1 and TP2 were equal to +3. The distance between the central carbon atoms of the given two guanidinium groups was chosen as the measure of distance between these groups. It was done so because the carbon atom is positioned approximately in the geometric center of the guanidinium moiety, the positive charge of which is delocalized over three nitrogen atoms. The distance values were statistically processed using our custom C++ script over the entire ensemble of approximately 100,000 conformations and the subensembles comprised of all obtained conformations, the energies of which did not exceed a certain cutoff value relative to the lowest energy conformation. A total of 5 subsensembles were created based on the cutoff values of 4, 4.5, 5, 6, and 7 kcal/mol, respectively. Conformational analysis was carried out using the RSC data center supercomputer system of the Ioffe Institute RAS, which made it possible to complete all calculations in 30 days.

### 4.6. Statistical Analysis

The data were analyzed with STATISTICA 10.0 (StatSoft, Inc. (2011). STATISTICA (data analysis software system), version 10. Tulsa, OK, USA) using the Student’s *t*-test and expressed as the mean value ± SEM. Statistical significance was set at *p* < 0.05.

## Figures and Tables

**Figure 1 ijms-23-05993-f001:**
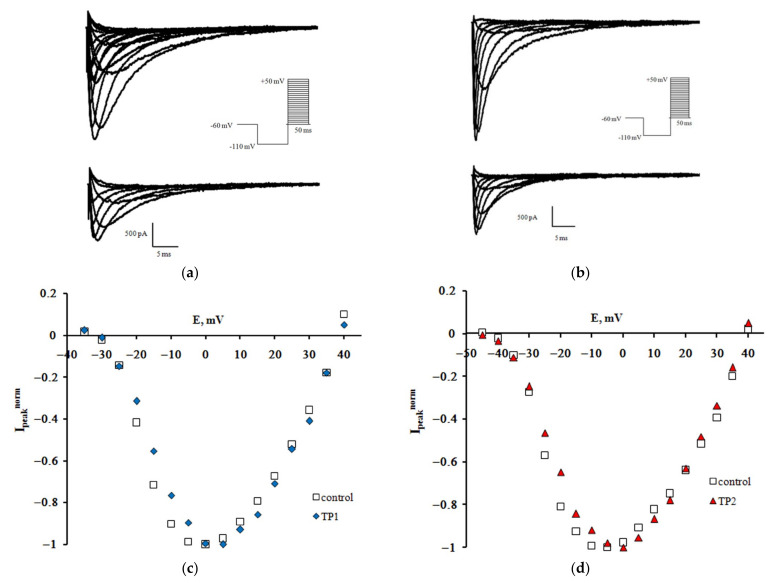
TP1 and TP2 effect on the Na_V_1.8 channel. Families of currents recorded before (top) and after (bottom) extracellular application of TP1 (**a**) and TP2 (**b**). Normalized peak current-voltage functions of the Na_V_1.8 channel in the control experiment and after application of TP1 (**c**) and TP2 (**d**). The holding potential of 500-ms duration was equal to −110 mV in all records. The leakage and capacitive currents were subtracted automatically. Voltage protocols are presented in the inserts.

**Figure 2 ijms-23-05993-f002:**
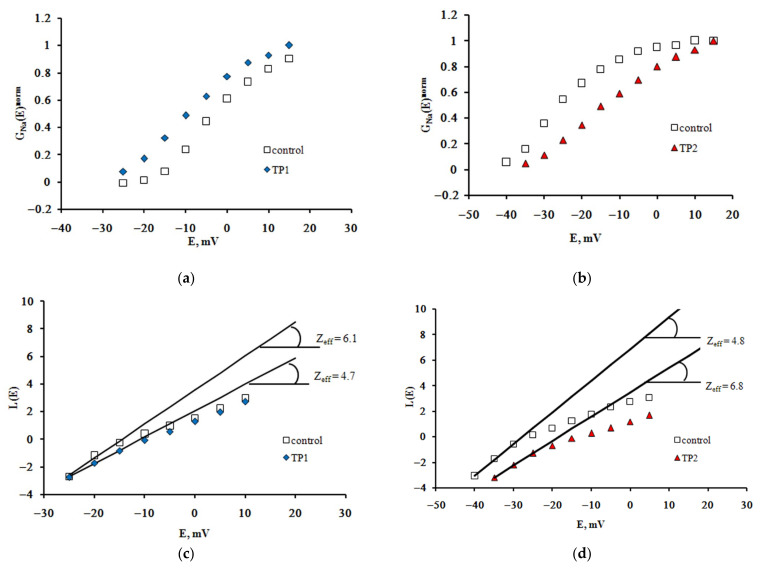
TP1 and TP2 modulate voltage sensitivity of the Na_V_1.8 channel activation gating device. Voltage dependence of the Na_V_1.8 channel chord conductance in the control experiment and after application of TP1 (**a**) and TP2 (**b**). The function G_Na_(E) was normalized, i.e., we plotted G_Na_(E)^norm^ = G_Na_(E)/G_Na_^max^(E), where G_Na_^max^(E) is the maximal value of G_Na_(E). Z_eff_ was evaluated from the tangents of the slopes of the asymptotes passing through the very first points of the L(E) function. Z_eff_ decreased significantly after application of TP1 (**c**) and TP2 (**d**).

**Figure 3 ijms-23-05993-f003:**
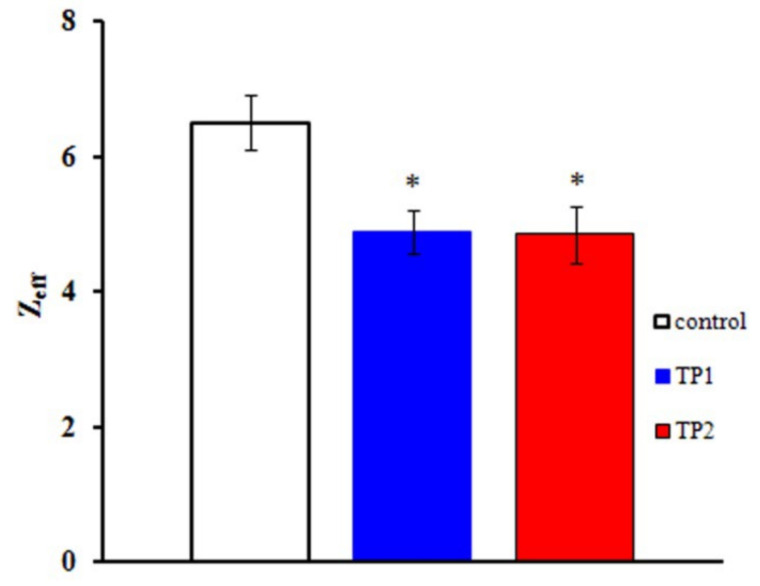
TP1 and TP2 decrease the effective charge of the Na_V_1.8 channel activation gating device. The Z_eff_ values of the Na_V_1.8 channel activation gating device after application of TP1 and TP2: control, Z_eff_ = 6.5 ± 0.4 (*n* = 23); effective charge after application of TP1, Z_eff_ = 4.9 ± 0.3 (*n* = 22); effective charge after application of TP2, Z_eff_ = 4.8 ± 0.4 (*n* = 19). Statistically significant differences between the control and experimental values are designated with asterisks (*p* < 0.05).

**Figure 4 ijms-23-05993-f004:**
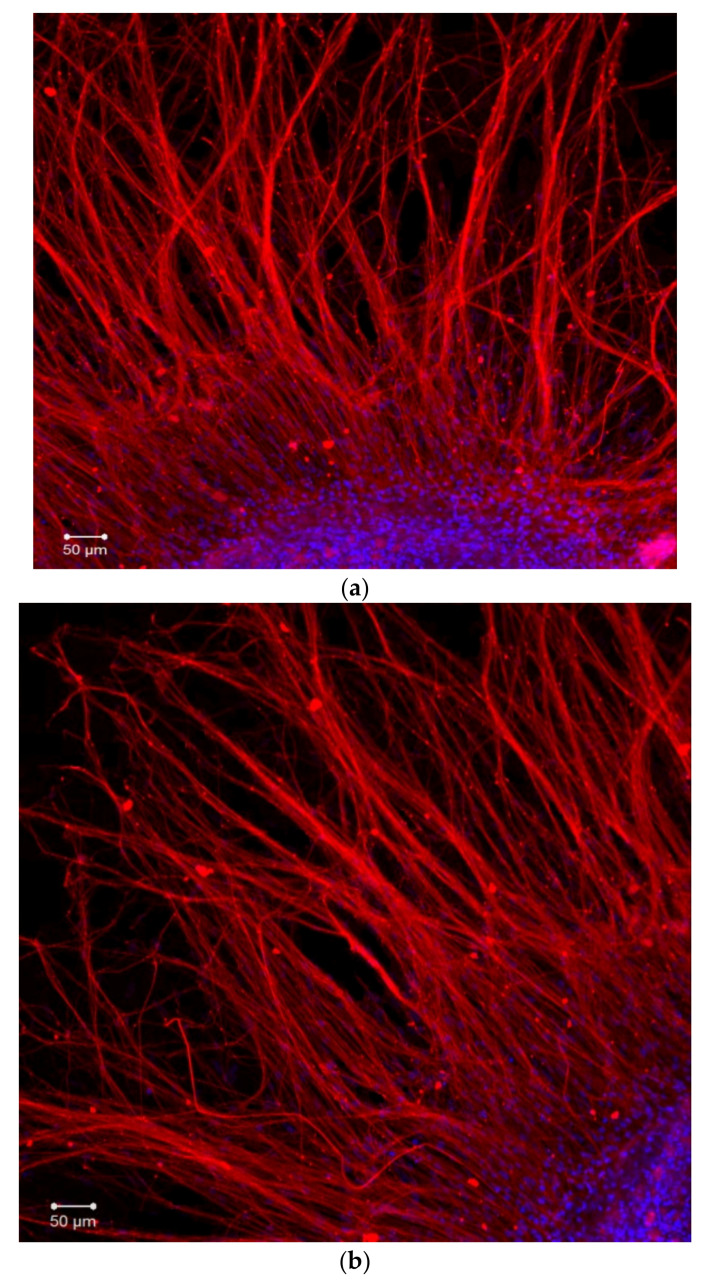
Fragments of DRG explant growth zone (third day of culturing). DRG explants were immunostained with anti-neurofilament 200 antibody (red). Nuclei were counterstained with DAPI (blue). Scale bar 50 μm. (**a**)—control, (**b**)—TP1 (100 nM), (**c**)—TP2 (100 nM).

**Figure 5 ijms-23-05993-f005:**
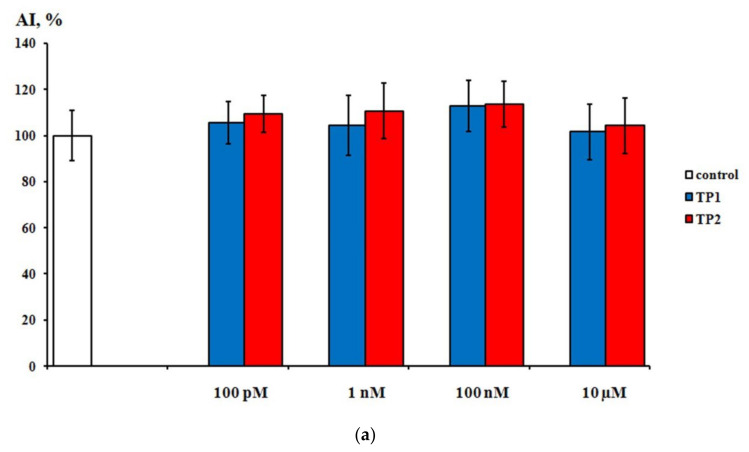
Effects of TP1 and TP2 on neurite growth in DRG explants. (**a**)—DRG area index values. The ordinate axis–area index (AI, %). Data are presented as mean ± SEM (*n* = 25 for each concentration; not significant, *p* > 0.5). (**b**)—DRG neurite length values. The ordinate axis–neurite length (%). Data are presented as mean ± SEM (*n* = 25 for each concentration; not significant, *p* > 0.6).

**Figure 6 ijms-23-05993-f006:**
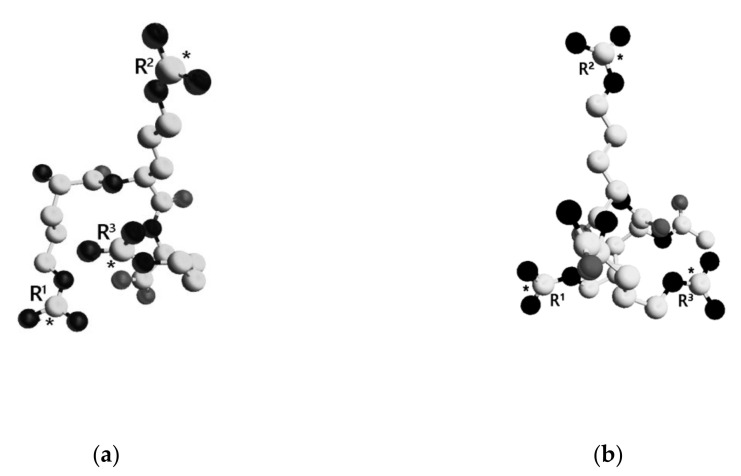
Spatial structure of the lowest energy conformations of TP1 and TP2. The dielectric constant ε = 10. Amino acid residues are numbered. Carbon atoms are presented as white spheres, oxygen atoms—gray, and nitrogen atoms—black. Carbon atoms of the guanidinium groups are marked with asterisks. Hydrogen atoms are not shown for clarity. (**a**)—TP1, (**b**)—TP2.

**Table 1 ijms-23-05993-t001:** Average distances between the guanidinium groups in TP1 and TP2.

Cutoff, kcal/mol	TP1				TP2			
	ε = 10		ε = 80		ε = 10		ε = 80	
	N_conf_	Distances, Å	N_conf_	Distances, Å	N_conf_	Distances, Å	N_conf_	Distances, Å
none	102,930	R^1^–R^2^ 10.6 ± 2.7 R^1^–R^3^ 11.2 ± 3.6 R^2^–R^3^ 10.6 ± 2.7	103,592	R^1^–R^2^ 10.5 ± 2.8 R^1^–R^3^ 11.3 ± 3.6 R^2^–R^3^ 10.6 ± 2.7	102,764	R^1^–R^2^ 10.7 ± 2.6 R^1^–R^3^ 11.1 ± 3.7 R^2^–R^3^ 11.0 ± 2.7	101,296	R^1^–R^2^ 11.1 ± 2.7 R^1^–R^3^ 11.3 ± 3.7 R^2^–R^3^ 11.0 ± 2.6
7	8856	R^1^–R^2^ 10.2 ± 2.5 R^1^–R^3^ 10.8 ± 3.5 R^2^–R^3^ 10.0 ± 2.4	8096	R^1^–R^2^ 10.2 ± 2.5 R^1^–R^3^ 10.8 ± 3.5 R^2^–R^3^ 9.9 ± 2.4	15,428	R^1^–R^2^ 10.6 ± 2.6 R^1^–R^3^ 9.6 ± 3.4 R^2^–R^3^ 10.9 ± 2.6	14,630	R^1^–R^2^ 11.2 ± 2.5 R^1^–R^3^ 9.9 ± 3.4 R^2^–R^3^ 10.8 ± 2.5
6	5270	R^1^–R^2^ 10.2 ± 2.5 R^1^–R^3^ 10.9 ± 3.5 R^2^–R^3^ 9.8 ± 2.4	4596	R^1^–R^2^ 10.2 ± 2.5 R^1^–R^3^ 10.8 ± 3.5 R^2^–R^3^ 9.8 ± 2.4	9798	R^1^–R^2^ 10.4 ± 2.6 R^1^–R^3^ 9.3 ± 3.3 R^2^–R^3^ 10.9 ± 2.5	9117	R^1^–R^2^ 11.0 ± 2.5 R^1^–R^3^ 9.5 ± 3.3 R^2^–R^3^ 10.8 ± 2.4
5	2463	R^1^–R^2^ 10.3 ± 2.4 R^1^–R^3^ 10.5 ± 3.6 R^2^–R^3^ 9.7 ± 2.4	1926	R^1^–R^2^ 10.3 ± 2.5 R^1^–R^3^ 10.2 ± 3.6 R^2^–R^3^ 9.7 ± 2.3	5197	R^1^–R^2^ 10.3 ± 2.6 R^1^–R^3^ 9.0 ± 3.0 R^2^–R^3^ 11.0 ± 2.4	4686	R^1^–R^2^ 10.9 ± 2.4 R^1^–R^3^ 9.2 ± 3.0 R^2^–R^3^ 10.8 ± 2.4
4.5	1438	R^1^–R^2^ 10.4 ± 2.4 R^1^–R^3^ 10.1 ± 3.6 R^2^–R^3^ 9.8 ± 2.4	1010	R^1^–R^2^ 10.3 ± 2.4 R^1^–R^3^ 9.6 ± 3.5 R^2^–R^3^ 9.7 ± 2.2	3498	R^1^–R^2^ 10.2 ± 2.6 R^1^–R^3^ 8.8 ± 2.8 R^2^–R^3^ 11.0 ± 2.4	3106	R^1^–R^2^ 10.8 ± 2.4 R^1^–R^3^ 9.0 ± 2.8 R^2^–R^3^ 10.9 ± 2.4
4	693	R^1^–R^2^ 10.4 ± 2.4 R^1^–R^3^ 9.4 ± 3.6 R^2^–R^3^ 9.7 ± 2.3	487	R^1^–R^2^ 10.3 ± 2.4 R^1^–R^3^ 8.8 ± 3.4 R^2^–R^3^ 9.6 ± 2.2	2247	R^1^–R^2^ 10.2 ± 2.6 R^1^–R^3^ 8.7 ± 2.7 R^2^–R^3^ 11.1 ± 2.4	2039	R^1^–R^2^ 10.7 ± 2.4 R^1^–R^3^ 9.0 ± 2.7 R^2^–R^3^ 11.0 ± 2.3

## Data Availability

Not applicable.

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
