# Peer review of "Arginine-Containing Tripeptides as Analgesic Substances: The Possible Mechanism of Ligand-Receptor Binding to the Slow Sodium Channel"

_ijms, 2022, doi:10.3390/ijms23115993_

Round 1

Reviewer 1 Report

General comments

Rogachevskii and colleagues present a paper entitled “Arginine-containing tripeptides as analgesic substances: the 2 possible mechanism of ligand-receptor binding to the slow 3 sodium channel”.

The authors have tested the effect of two short arginine-containing tripeptides, H-Arg-Arg-Arg-OH (TP1) and 13 Ac-Arg-Arg-Arg-NH2 (TP2) on Nav1.8 channel of DRG primary cultures, by using patch-clamp recording. They demonstrated that both TP1 and TP2 tripeptides modulate Nav1.8 channels and in particular decrease Zeff at nanomolar concentration. In addition, demonstrated that the compounds do not effect neurite growth even if a more detailed analyses would be necessary.

In my opinion, this article responds to the scope of International Journal of Molecular Sciences and should be of interest and useful for its readership. Anyway, the figures are a little confusing and the manuscript requires major revision, to be clearer and more exhaustive for the reader.

  1. For what is peak current normalized? From single traces seems that TP1 and TP2 decrease current amplitude, but this effect is not apparent from figure 1c and d.
  2. In figures 1c-d and 2 is it show a single experiment? if that's the case, please insert the average data of Ipeak and GNa(E).
  3. Please, insert the same axis scales in figures 2c and d. moreover, TP2 seems to be more effective than TP1 on Zeff parameter.
  4. For neurite outgrowth, authors measure only area index, but this is a rough parameter. Please also measure parameters as “total neurite length”.
  5. Please, add more reference in introduction, discussion, but especially in material and methods (a time of enzymatic treatment of 2-5 minutes is too short)

Author Response

  1. 1. For what is peak current normalized? From single traces seems that TP1 and TP2 decrease current amplitude, but this effect is not apparent from figure 1c and d.

Response: done. Lines 224-290
2. In figures 1c-d and 2 is it show a single experiment? if that's the case, please insert the average data of Ipeak and GNa(E).

Response: done. Lines 115-117, 224-290
3. Please, insert the same axis scales in figures 2c and d. moreover, TP2 seems to be more effective than TP1 on Zeff parameter.

Response: done. Lines 118-127

  1. For neurite outgrowth, authors measure only area index, but this is a rough parameter. Please also measure parameters as “total neurite length”.

Response: done. Lines 151-153, 171, 173-176, 481-489
5. Please, add more reference in introduction, discussion, but especially in material and methods (a time of enzymatic treatment of 2-5 minutes is too short)

Response: done. Lines 372-373, 586-587

Reviewer 2 Report

In this manuscript, the authors investigated the analgesic effects of two arginine-containing tripeptides, TP1 and TP2. Both tripeptides were shown to modulate the voltage dependence of Nav1.8 channel activation using patch clamp. In addition, TP1 and TP2 didn’t affect the neurite growth of dorsal root ganglions. The data seems convincing, and the conclusions seem appropriate. However, there are some concerns that need to address.   

Major concerns:

  1. The effects of TP1 and TP2 on the inactivation of Nav1.8 channels are needed to support the conclusion that TP1 and TP2 modulate the voltage sensitivity of Nav1.8 channels. Because both activation and inactivation are important gating properties of Nav1.8 channels.
  2. Should the authors mention the decrease of Ipeak in figure 1? In other words, the current amplitude at all voltages was reduced by TP1 and TP2, any mechanism?
  3. Figure 2a,b, why the voltage dependence of Nav1.8 is negatively shifted by TP1 while positively shifted by TP2?

Minor concerns:

  1. Is it possible to show the P-value in Figures 3 and 5?
  2. The voltage protocol is recommended to show in figure 1.
  3. English editing is highly recommended. Some sentences are extremely long and hard to read. For example, Line 266, 18, 176 etc.

Author Response

Major concerns: 

  1. The effects of TP1 and TP2 on the inactivation of Nav1.8 channels are needed to support the conclusion that TP1 and TP2 modulate the voltage sensitivity of Nav1.8 channels. Because both activation and inactivation are important gating properties of Nav1.8 channels.

Response: done. Lines 224-290
2. Should the authors mention the decrease of Ipeak in figure 1? In other words, the current amplitude at all voltages was reduced by TP1 and TP2, any mechanism?

Response: done. Lines 224-290
3. Figure 2a,b, why the voltage dependence of Nav1.8 is negatively shifted by TP1 while positively shifted by TP2?

Response: done. Lines 118-127, 434-434

Minor concerns:

  1. Is it possible to show the P-value in Figures 3 and 5?

Response: done. Lines 137-141, 173-176

  1. The voltage protocol is recommended to show in figure 1.

Response: done. Lines 86-88

  1. English editing is highly recommended. Some sentences are extremely long and hard to read. For example, Line 266, 18, 176 etc.

Response: done. Lines 37-42, 73-77, 210-211, 302-306, 310-314, 351-357, 510-514

Round 2

Reviewer 1 Report

The authors have been answered correctly to the revision.
In my opinion, now the manuscript is more clearly and can be accepted.

Reviewer 2 Report

The authors have addressed all my concerns.